# Global genetic differentiation of complex traits shaped by natural selection in humans

Jing Guo[1], Yang Wu [1], Zhihong Zhu[1], Zhili Zheng[1,2], Maciej Trzaskowski[1], Jian Zeng[1], Matthew R. Robinson [1,3], Peter M. Visscher [1,4] & Jian Yang [1,4]

There are mean differences in complex traits among global human populations. We hypothesize that part of the phenotypic differentiation is due to natural selection. To address this hypothesis, we assess the differentiation in allele frequencies of trait-associated SNPs among African, Eastern Asian, and European populations for ten complex traits using data of large sample size (up to ~405,000). We show that SNPs associated with height ($P = 2.46 \times 10^{-5}$), waist-to-hip ratio ($P = 2.77 \times 10^{-4}$), and schizophrenia ($P = 3.96 \times 10^{-5}$) are significantly more differentiated among populations than matched "control" SNPs, suggesting that these trait-associated SNPs have undergone natural selection. We further find that SNPs associated with height ($P = 2.01 \times 10^{-6}$) and schizophrenia ($P = 5.16 \times 10^{-18}$) show significantly higher variance in linkage disequilibrium (LD) scores across populations than control SNPs. Our results support the hypothesis that natural selection has shaped the genetic differentiation of complex traits, such as height and schizophrenia, among worldwide populations.

[1] Institute for Molecular Bioscience, The University of Queensland, Brisbane, QLD 4072, Australia. [2] The Eye Hospital, School of Ophthalmology and Optometry, Wenzhou Medical University, 325027 Zhejiang, China. [3] Department of Computational Biology, University of Lausanne, 1011 Lausanne, Switzerland. [4] Queensland Brain Institute, The University of Queensland, Brisbane, QLD 4072, Australia. Correspondence and requests for materials should be addressed to J.Y. (email: jian.yang@uq.edu.au)

Many human complex traits, including quantitative traits (e.g., height[1]) and complex disorders (e.g., cardiovascular diseases[2,3]), are substantially differentiated among worldwide populations. For example, the mean height in Northern Hemisphere populations generally increases with latitude[1,4]. European Americans have a lower body mass index (BMI) (~1.3 kg/m$^2$) than African Americans but a higher BMI (1.9–3.2 kg/m$^2$) than Asians, such as Chinese, Indonesians, and Thais for the same body fat percentage[5,6]. For the mortality rates associated with ischemic heart disease in the UK, African Caribbeans are at a lower risk while South Asians are at a higher risk than Europeans[7]. While environmental factors certainly play a role, since most complex traits have a genetic component, the question is whether or not the phenotypic differentiation is partly due to genetic differentiation and, if so, whether the genetic differentiation is a consequence of genetic drift or natural selection. There has been evidence suggesting that natural selection has caused genetic differentiation among worldwide populations[8–10] and the signals of selection are enriched in specific parts of the genome[8,11]. However, it is not straightforward to investigate whether the signals of natural selection are enriched at genetic variants associated with a complex trait for two main reasons. First, because of the polygenic nature of most complex traits[12], the signals of genetic differentiation are usually diluted among many trait-associated loci, each having a small effect too weak to be detected using methods that target a complete selective sweep[13–15]. Second, genetic differentiation can be masked by environmental factors, thereby increasing the difficulty of directly detecting and quantifying the polygenic selection signals[4,16].

Over the past decade, genome-wide association studies (GWAS) have identified thousands of single nucleotide polymorphisms (SNPs) associated with a number of traits in humans[17]. These findings have provided critical knowledge for understanding the polygenic architecture of human complex traits[18] or detecting the signature of natural selection[4,16,19,20]. The large amount of GWAS data available in the public domain also allows researchers to address the question whether genetic variants associated with a complex trait have been under natural selection. For example, utilizing data from GWAS of large sample size, recent studies have shown that genetic variants associated with human height have been under directional selection in European populations[4,19,21]. In this study, we seek to address the question whether the differentiation in allele frequency and linkage disequilibrium (LD) of variants associated with a complex trait among global populations is shaped by natural selection. We first test if allele frequencies of the trait-associated SNPs are more differentiated across African, East Asian, and European populations than expected under genetic drift for 10 complex traits (Supplementary Table 1) utilizing summary-level data from published GWAS with large sample sizes (n = up to ~405,000). These traits include three quantitative traits (adult height, BMI, and waist-to-hip ratio adjusted by BMI (WHRadjBMI)), two common diseases and related biochemical traits (coronary artery disease (CAD) and type II diabetes (T2D) and high-density lipoprotein (HDL) cholesterol and low-density lipoprotein (LDL) cholesterol), two neurological/psychiatric disorders (Alzheimer's disease (AD) and schizophrenia (SCZ)), and one behavioral trait (educational attainment years (EAY)). For the traits for which the associated SNPs are significantly more differentiated among the global populations than expected under drift, we then measure the direction of genetic differentiation using polygenic scoring[4,16] or the mean frequency of the trait-increasing alleles, and compare it with the observed direction of phenotypic differentiation in the three populations. Moreover, previous studies have shown that both strong and soft selective sweeps can alter linkage disequilibrium (LD) between a genetic variant and its surrounding variants[22–25]. If a trait-associated SNP has been under selection, one would also expect to see a population differentiation of the LD score of this SNP with its surrounding SNPs, more than expected under a drift model, where the LD score is defined as the sum of the LD $r^2$ between the focal SNP and all nearby SNPs in a 10 Mb window[26,27]. In this context, we further test whether the trait-associated SNPs show greater differences in LD across populations than the control SNPs.

## Results

**Enrichment of $F_{ST}$ in trait-associated SNPs.** If the genetic loci associated with a complex trait have been under natural selection, an excess of among-population differentiation will be observed in the frequencies of the trait-associated alleles compared with what is expected under a drift model[28]. We used Wright's fixation index ($F_{ST}$) to measure the extent to which a particular SNP varied in allele frequency among three worldwide populations. We focused only on common variants (minor allele frequency, MAF > 0.01) because rare variants were not available in most GWAS summary data used in this study. The $F_{ST}$ values of the SNPs were calculated from unrelated individuals (SNP-derived genetic relatedness < 0.05) of European (EUR, n = 1099), African (AFR, n = 1099), and East Asian (EAS, n = 1099) ancestry from the Genetic Epidemiology Research on Adult Health and Aging (GERA) cohort after quality controls (QC) (Methods; Supplementary Table 2). We selected a list of nearly independent SNPs (LD $r^2$ < 0.01) associated with each trait using PLINK[29] clumping analysis (Methods) of the summary statistics from the latest meta-analysis of GWAS (Supplementary Table 1) and a set of "control" SNPs randomly sampled from the genome with MAF and LD scores matched with those of the associated SNPs (Methods). All GWAS summary statistics were from studies on individuals of European ancestry. We tested whether the mean $F_{ST}$ value of the trait-associated SNPs was significantly higher than that of the control SNPs, a method we call the $F_{ST}$ enrichment test (Methods), similar to the approach used in Zhang et al.[30]. We demonstrate by simulation (Methods; Supplementary Table 3) that there was no inflation in the test-statistics of the method under the null model of genetic drift (Supplementary Fig. 1a) and that selecting associated SNPs at a clumping $P$-value threshold of $5 \times 10^{-6}$ provided higher detection power than at $P < 5 \times 10^{-8}$ under the alternative model (Supplementary Fig. 1b). We performed the $F_{ST}$ enrichment analysis for each of the 10 traits using the trait-associated SNPs clumped at $P < 5 \times 10^{-6}$ with $F_{ST}$ values computed from the three populations in GERA. We found that the mean $F_{ST}$ values for the trait-associated loci for height ($P = 2.46 \times 10^{-5}$), WHRadjBMI ($P = 2.77 \times 10^{-4}$), and SCZ ($P = 3.96 \times 10^{-5}$) were significantly higher than those of the control SNPs after Bonferroni correction for multiple tests (Table 1), indicating that the genetic loci associated with these traits have been under natural selection. We further confirmed the results using $F_{ST}$ values calculated from the 1000 Genome Project (1000G; unrelated EUR n = 494, AFR n = 591, and EAS n = 491; Supplementary Table 2). The results remained significant for height ($P = 4.93 \times 10^{-6}$), WHRadjBMI ($P = 2.62 \times 10^{-3}$), and SCZ ($P = 6.83 \times 10^{-5}$) correcting for multiple tests (Table 1 and Fig. 1).

**Direction of genetic differentiation.** The analyses above showed that the genetic variants associated with height, WHRadjBMI, and SCZ are more differentiated than expected by random drift. However, these analyses did not show the direction of genetic differentiation. To demonstrate the direction of differentiation, we used the SNPs clumped at $P < 5 \times 10^{-6}$ (see above) to compute

**Table 1 Enrichment test of the differentiation of trait-associated SNPs (clumped at $P < 5 \times 10^{-6}$) in allele frequency against the control SNPs among the three populations in GERA and 1000G**

| Trait | GERA | | | 1000G | | |
|---|---|---|---|---|---|---|
| | Number of SNPs | Mean $F_{ST}$ | P-value | Number of SNPs | Mean $F_{ST}$ | P-value |
| **Height** | 1044 | 0.090 | **$2.46 \times 10^{-5}$** | 1099 | 0.140 | **$4.93 \times 10^{-6}$** |
| BMI | 157 | 0.093 | 0.038 | 179 | 0.136 | 0.344 |
| **WHRadjBMI** | 82 | 0.109 | **$2.77 \times 10^{-4}$** | 92 | 0.158 | **$2.62 \times 10^{-3}$** |
| HDL | 175 | 0.093 | 0.017 | 181 | 0.128 | 0.672 |
| LDL | 143 | 0.079 | 0.804 | 139 | 0.115 | 0.421 |
| EAY | 312 | 0.083 | 0.359 | 328 | 0.139 | 0.021 |
| AD | 43 | 0.070 | 0.614 | 46 | 0.108 | 0.545 |
| CAD | 101 | 0.079 | 0.731 | 102 | 0.119 | 0.892 |
| **SCZ** | 334 | 0.093 | **$3.96 \times 10^{-5}$** | 337 | 0.145 | **$6.83 \times 10^{-5}$** |
| T2D | 40 | 0.082 | 0.885 | 38 | 0.129 | 0.772 |

Bold values represent the results that are significant in GERA and replicated in 1000G, correcting for multiple tests

**Fig. 1** Mean $F_{ST}$ values of the associated SNPs across 1000G populations against the null distribution for height, WHRadjBMI and SCZ. The red dashed line represents the mean $F_{ST}$ of the trait-associated SNPs clumped at $P < 5 \times 10^{-6}$. The histogram represents the distribution of mean $F_{ST}$ values of the sets of control SNPs. WHRadjBMI waist-to-hip ratio adjusted by BMI, SCZ schizophrenia

a polygenic risk score (PRS)[4,16,29] for each individual in 1000G for height, WHRadjBMI, and SCZ. We then estimated the deviation of the mean PRS of each population from the overall mean in standard deviation (s.d.) units (Methods). The deviation is expected to be zero under drift as demonstrated by the null distribution computed from the 10,000 control SNP sets (Fig. 2). The results showed that the mean PRS for height in the EUR subjects was higher than that in the AFR and EAS subjects (Fig. 2), consistent with the observed mean phenotypic differences between the populations[1]. For SCZ, the mean PRS in AFR was higher than that in EUR (Fig. 2), in line with the results from recent studies suggesting that SCZ is more prevalent in people of AFR ancestry than EUR ancestry and Asians[31–33]. For WHRadjBMI, EAS showed a higher mean PRS than both AFR and EUR (Fig. 2). WHRadjBMI measures the fat distribution at the abdomen region after adjusting for BMI to exclude the influence of overall adiposity. Previous studies have reported that after correcting for age, gender and BMI, Asian Americans tend to have a higher accumulation of excess visceral adipose tissue (VAT) than European Americans[34], whereas African Americans have a lower VAT than European Americans[35,36]. We repeated the PRS analysis in GERA and found that the results were consistent with those in 1000G (Supplementary Fig. 2). It should be noted that we first used the $F_{ST}$ enrichment analysis to seek for evidence that the SNPs associated with a trait have been under natural selection, and then used the PRS analysis to illustrate the direction of selection without repeating the significance test. An observed differentiation of PRS values among populations alone is not evidence for selection unless it is more than expected under drift.

The PRS analysis used the effect sizes of SNPs estimated from GWAS samples of EUR ancestry. Strong genetic heterogeneity between EUR and non-EUR (e.g., if the effect sizes in EUR are different from those in non-EUR populations) could possibly bias PRS prediction in non-EUR populations[37,38]. To avoid this potential bias, we examined the direction of genetic differentiation by directly comparing the difference in mean frequency of the trait-increasing alleles (fTIAs) across SNPs (clumped at $P < 5 \times 10^{-6}$) between two populations because in comparison with the magnitude, the direction of SNP effect estimated from GWAS is less prone to bias due to population structure[37]. We also calculated the difference in fTIA between the two populations for the control SNPs. Note that TIA of a control SNP simply means the allele for which the estimated SNP effect is positive in the GWAS summary data. Under a drift model, fTIA is expected to be 0.5 and the difference in fTIA values between two populations is expected to be zero. The results were consistent with those from the PRS analysis (Figs. 2, 3). On average, the height-increasing alleles were more frequent in EUR than in EAS, SCZ risk alleles were more frequent in ARFs than in EUR, and WHRadjBMI-increasing alleles were more frequent in EAS than in EUR (see Fig. 3 for the results from 1000G and Supplementary Fig. 3 for the results from GERA). Of note, the mean fTIA for the control SNPs was not zero, especially for height and SCZ, which was likely because both height and SCZ are highly polygenic[39,40] and some of the control SNP could be in LD with the causal variant(s) by chance. This result also implies that SNPs other than those selected at low-association P-values have also been under polygenic selection[10,14].

**LD pattern of complex trait loci altered by selection**. Because our findings indicated that natural selection has shaped the frequencies of the trait-associated alleles, we then sought to test whether selection has also differentiated the LD pattern at the trait-associated genomic loci among populations more than expected under drift[22–25]. If this is the case, we should expect to see larger among-population difference in LD score at the trait-associated SNPs than at the control SNPs. We first computed the LD score of each SNP in the unrelated individuals in GERA and 1000G, respectively. We then calculated the coefficient of variation of the LD score (LDCV) across the AFR, EAS, and EUR populations for each SNP (Methods), and tested whether the mean LDCV of the trait-associated SNPs (clumped at $P < 5 \times 10^{-6}$) significantly deviates from that of the control SNPs, a procedure we call the LDCV enrichment analysis. Note that we used coefficient of variation rather than variance because SNPs with higher LD scores tend to have larger between-population variation, a mean-variance relationship. Using the LDCV from GERA, we found a significant excess of LDCV for the trait-associated SNPs over the control SNPs for height ($P = 2.01 \times 10^{-6}$), EAY ($P = 2.37 \times 10^{-8}$), and SCZ ($P = 5.16 \times 10^{-18}$) after Bonferroni correction for multiple tests (Table 2). The results for height ($P = 1.99 \times 10^{-4}$) and SCZ ($P = 2.74 \times 10^{-8}$) remained significant (after correcting for multiple tests) when using the LDCV from 1000G (Table 2 and Supplementary Fig. 4). There was no significant correlation between $F_{ST}$ and LDCV for either height or

SCZ, suggesting that the between-population differentiation in LD were not confounded by the differentiation in $F_{ST}$ (Supplementary Fig. 5). Together, these results suggest that natural selection has altered both the frequency and LD properties of the SNPs associated with height and SCZ.

**Discussion**

This study sought to address whether natural selection has differentiated allele frequencies and LD scores of complex traits associated SNPs among worldwide populations more than expected under drift. To this end, we established a strategy to test whether the among-population variation in allele frequency or LD score at the trait-associated SNPs is significantly higher than that at MAF-matched and LD-matched control SNPs. We detected significant signals in allele frequencies in the GERA populations for height ($P = 2.46 \times 10^{-5}$), WHRadjBMI ($P = 2.77 \times 10^{-4}$), and SCZ ($P = 3.96 \times 10^{-5}$) (Table 1) and significant signals in LD for height ($P = 2.01 \times 10^{-6}$) and SCZ ($P = 5.16 \times 10^{-18}$) (Table 2). There are two plausible models that are compatible with the observed results: 1) the trait itself has undergone natural selection; 2) variants affecting the trait have been under selection because of their pleiotropic effects on fitness, and variants with larger effects on the trait tend to be more differentiated via such pleiotropic effects[41].

Height is a classic complex trait. Previous studies have shown that height-associated SNPs have been under natural selection and that this might contribute to the mean height differences among EUR populations[4,16,19–21]. Our results suggest that the phenotypic differences in height among AFR, EAS, and EUR populations are also partially due to natural selection on height-associated SNPs (Fig. 2 and Supplementary Fig. 2). The mean PRS of AFR was higher than that of EAS but lower than that of EUR, consistent with the observed phenotypic differences in mean height[1]. Waist-to-hip ratio is an anthropometric index of abdominal obesity. Previous studies have shown that after adjusting for BMI, European Americans tend to have a larger amount of visceral fat on average than African Americans[35,36] but a lower amount than those of Asian descent[6,34]. This finding could be explained by the enrichment of the WHRadjBMI-increasing alleles in EAS as shown in our results; we did not have enough power to detect difference between EUR and AFR (Fig. 2 and Supplementary Fig. 2). It is worth noting that, compared with height, direct measurements of WHRadjBMI are not widely available for worldwide populations and are less consistent across studies, which may be related to insufficient sample sizes[42,43] or the influence of environmental factors[44]. For SCZ, the enrichment of risk alleles in AFR (Fig. 2 and Supplementary Fig. 2) identified by our analyses appears to support the reported

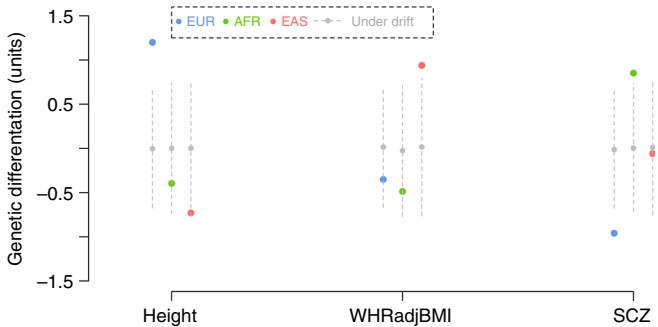

**Fig. 2** Direction of genetic differentiation for height, WHRadjBMI and SCZ in the 1000G populations. The colored dot represents the estimated deviation (in s.d. units) of the mean PRS based on the trait-associated SNPs clumped at $P < 5 \times 10^{-6}$ of a population from the overall mean across populations. The gray dot represents the mean of mean PRS values of 10,000 sets of control SNPs, with the gray dashed line indicating the 95% confidence interval of the distribution of mean PRS values. WHRadjBMI, waist-to-hip ratio adjusted by BMI; SCZ schizophrenia, EUR European, AFR African, EAS East Asian

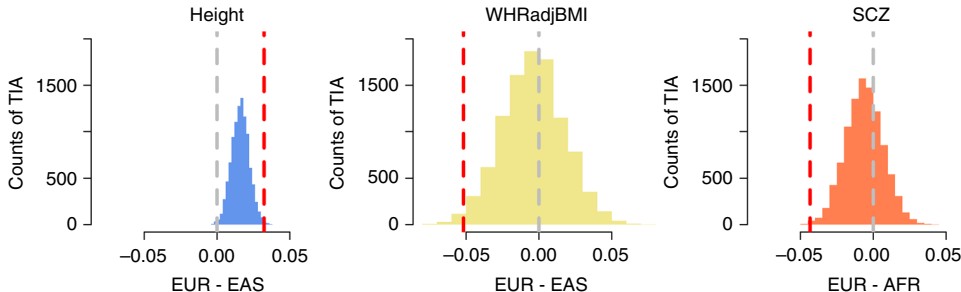

**Fig. 3** Mean difference in frequencies of the trait-increasing alleles between two 1000G populations for height, WHRadjBMI and SCZ. The red dashed line represents the mean difference in fTIA of the trait-associated SNPs clumped at $P < 5 \times 10^{-6}$. The histogram represents the distribution of the difference in fTIA for the control SNPs. The gray dashed line represents the expected difference in fTIA (i.e., 0) under genetic drift. WHRadjBMI, waist-to-hip ratio adjusted by BMI; SCZ schizophrenia, EUR European, AFR African, EAS East Asian

**Table 2 Enrichment test of the differentiation of trait-associated SNPs (clumped at $P < 5 \times 10^{-6}$) in the LD pattern against the control SNPs among the three populations in GERA and 1000G**

| Trait | GERA | | | | 1000G | | | |
|---|---|---|---|---|---|---|---|---|
| | Associated SNPs | | Control SNPs | P-value | Associated SNPs | | Control SNPs | P-value |
| | Number of SNPs | Mean LDCV | Mean LDCV (s.d.) | | Number of SNPs | Mean LDCV | Mean LDCV (s.d.) | |
| **Height** | 1044 | 0.401 | 0.376 (0.005) | $2.01 \times 10^{-6}$ | 1,099 | 0.390 | 0.373 (0.005) | $1.99 \times 10^{-4}$ |
| BMI | 157 | 0.402 | 0.375 (0.013) | 0.037 | 179 | 0.400 | 0.38 (0.011) | 0.089 |
| WHRadjBMI | 82 | 0.362 | 0.373 (0.019) | 0.539 | 92 | 0.369 | 0.368 (0.016) | 0.944 |
| HDL | 175 | 0.411 | 0.381 (0.014) | 0.025 | 181 | 0.373 | 0.371 (0.012) | 0.861 |
| LDL | 143 | 0.393 | 0.383 (0.015) | 0.521 | 139 | 0.378 | 0.37 (0.013) | 0.52 |
| EAY | 312 | 0.429 | 0.374 (0.01) | $2.37 \times 10^{-8}$ | 328 | 0.398 | 0.374 (0.008) | $5.2 \times 10^{-3}$ |
| AD | 43 | 0.413 | 0.388 (0.029) | 0.397 | 46 | 0.407 | 0.367 (0.024) | 0.093 |
| CAD | 101 | 0.402 | 0.383 (0.018) | 0.289 | 102 | 0.389 | 0.367 (0.016) | 0.164 |
| **SCZ** | 334 | 0.465 | 0.38 (0.01) | $5.16 \times 10^{-18}$ | 337 | 0.420 | 0.374 (0.008) | $2.74 \times 10^{-8}$ |
| T2D | 40 | 0.436 | 0.373 (0.028) | 0.025 | 38 | 0.430 | 0.368 (0.024) | $8.97 \times 10^{-3}$ |

Bold values represent the results that are significant in GERA and replicated in 1000G, correcting for multiple tests

discrepancy in the prevalence of SCZ between African Americans and European Americans[31,32]. Moreover, Fearon et al.[33] found that, among the ethnic minority groups in the UK, the incidence rate ratios (IRRs) for SCZ (adjusted by age and sex) in individuals of Chinese descent (IRR = 3.5) is higher than non-British Whites (IRR=2.5) and lower than individuals of African descent (IRR = 9.1 for African-Caribbean and 5.8 for Black African), consistent with our findings. However, the diagnosis of SCZ can be potentially biased by non-genetic factors, such as socioeconomic status and access to hospitalization[31].

The trait-associated SNPs used to detect signatures of natural selection were ascertained from EUR-based meta-analyses of GWAS. Such ascertainment might be biased because $F_{ST}$ is a function of MAF and the MAF or LD properties of the trait-associated SNPs could be different from that of the non-associated SNPs[26,45]. For example, SNPs with higher MAF in EUR tend to have higher power to be detected at a certain significance level, resulting in a difference in mean $F_{ST}$ even in the absence of natural selection. This potential bias can be mitigated by matching the control SNPs with the associated SNPs by MAF and LD score. Via simulations, we confirmed that inflation did not occur in the test statistics under the null model (Supplementary Fig. 1a). Our $F_{ST}$ enrichment results are unlikely to be driven by the EUR bias because among the traits that showed significant signal in the $F_{ST}$ enrichment test (Table 1) only height showed higher mean fTIA in EUR than non-EUR (mean fTIA was higher in EAS for WHRadjBMI and higher in AFR for SCZ compared to EUR) (Fig. 3). Moreover, we did not observe a correlation between the worldwide $F_{ST}$ (using the AFR, EAS and EUR samples in 1000G) and EUR $F_{ST}$ (using the CEPH, Finnish, British, Spanish, and Tuscan samples in 1000G-EUR) for the trait-associated SNPs that show evidence of selection (Supplementary Fig. 6). This result suggests that the differentiation of PRS in global populations is unlikely to be confounded by the biases in the estimated SNP effects due to population stratification in EUR. In addition, genetic drift that takes place during particular demographic events (e.g., population bottleneck or expansion in Europeans) would result in a difference in the frequency spectrum between the causal variants and neutral variants[46]. The difference, however, was very unlikely lead to a bias in our results because there was no difference in MAF between the trait-associated SNPs and the matched control SNPs. It is confirmed by simulation that there was no inflation in the $F_{ST}$ enrichment test-statistics when the

causal variants were simulated to have lower MAF than null SNPs in the absence of selection (A and B in Supplementary Table 4 and Supplementary Fig. 7a). We further demonstrated by simulation that the $F_{ST}$ enrichment analysis method was robust to different levels of heritability (C and D in Supplementary Table 4 and Supplementary Fig. 7b), different degrees of LD between causal variants (Supplementary Figs. 8 and 9), different strategies of sampling control SNPs (matching the control SNPs with the trait-associated SNPs by only MAF computed from EUR or by both MAF and LD scores computed from AFR; Supplementary Fig. 10), or whether the causal variants were included in the analysis (Supplementary Fig. 11). We also applied different strategies of matching control SNPs to the analyses of real data and observed little differences in results (Supplementary Table 5). In addition, we compared the variance of $F_{ST}$ values of a set of associated SNPs with a random set of control SNPs (with MAF and LD matched) across 100 independently simulated traits and did not observe a significant difference in variance of $F_{ST}$ values between the trait-associated SNPs and control SNPs, regardless of whether the traits were simulated based on a single variant or multiple variants at each of the 1,000 causal loci (Supplementary Fig. 12).

We observed from the LD score calculation in GERA that there were three regions (on chromosomes 6, 11 and 17) presenting extremely large LD scores (Supplementary Fig. 13). The chr6 region (Supplementary Fig. 14a) harbors genes that encode the major histocompatibility complex (MHC; hg19 chr6:28,477,797-33,448,354), a well-known protein complex that is essential for the immune response. It is not surprising that the MHC locus has been under selection[47]. The chr17 region (Supplementary Fig. 14b) harbors an inversion (hg16 chr17:44.1-45.0 Mb) that almost exclusively occurs in EUR and has been shown to be under positive selection in Icelandic females[48]. The chr11 region (Supplementary Fig. 14c) stretches over the centromere with a length >10 Mb, and it contains multiple polymorphic inversions[49]. More than half of the genes contained in this region (48–60 Mb; Supplementary Fig. 14d) are olfactory receptor genes (ORs) (186 ORs/308 genes)[50]. This gene family has been found to show the greater evolutionary acceleration between humans and chimpanzees than other functional gene classes, such as nuclear transport and reproduction[51]. Moreover, this region included a large number of nearly independent SNPs that show pleiotropic effects on HDL cholesterol and height (Supplementary Fig. 14d).

Nevertheless, several limitations were associated with this study. First, the $F_{ST}$ enrichment test is underpowered for highly polygenic traits because some of the control SNPs might be in LD with the causal variants by chance under a polygenic model (as demonstrated by the deviation of fTIA differences in the control SNPs from the expected values for height and SCZ; Fig. 3). Fortunately, the loss of power was remedied by the use of data from studies with very large sample sizes (Supplementary Table 1). Second, the GWAS summary statistics used in this study were generated from EUR samples (see the discussion of EUR bias above). This is because non-EUR GWAS of large sample size (on a similar scale as the sample sizes of the EUR GWAS used in this study) are not available for most complex traits. If there is genetic heterogeneity between EUR and non-EUR populations, the PRS computed in AFR and EAS based on SNP effects estimated from EUR studies will be biased[37,38]. We showed that this potential bias could be mitigated by the fTIA analysis, which ignores the magnitude of estimated SNP effects (Fig. 3 and Supplementary Fig. 3). In fact, the results from the fTIA analysis (Fig. 3 and Supplementary Fig. 3) were largely consistent with those from the PRS analysis (Fig. 2 and Supplementary Fig. 2), suggesting that the direction of genetic differentiation as indicated by the PRS for height, WHRadjBMI, or SCZ is unlikely to be substantially biased by the between-population differences in SNP effects[52]. In addition, our result that the EUR $F_{ST}$ was almost independent of the global $F_{ST}$ (Supplementary Fig. 6) implies that the mean values of PRS in non-EUR samples were not biased by possible confounding in the estimated SNP effects owing to population stratification in EUR. Third, because we tested the mean $F_{ST}$ (or LDCV) across all the associated loci against that of the control SNPs, we could not distinguish whether the population differentiation of mean $F_{ST}$ (or LDCV) at the trait-associated variants (more than expected by drift) is due to natural selection on different loci, the same loci but different alleles, or the same alleles but different levels of selection pressure in different populations. Fourth, regarding to the type of selection on the trait-associated variants, our results seem to indicate that the excess of genetic differentiation at the trait-associated SNPs is a consequence of local selection (different alleles are favored in different populations). However, we cannot rule out the possibility that the increase in $F_{ST}$ (or LDCV) at the trait-associated SNPs is due to background selection[53] (i.e., those SNPs are in LD with causal variants under negative selection). This is confirmed by forward simulation using SLiM[54] (Methods). The simulation result shows that background selection reduces genetic diversity and increases between-population differentiation at genetic variants in LD with the variants under negative selection (Supplementary Fig. 15). Last, we used the $F_{ST}$ and LDCV enrichment analyses to assess the excess of population genetic differentiation at the trait-associated loci as a means to detect signature of natural selection. These analyses, however, cannot determine when the selection occurred in the history of human evolution and whether there are other types of natural selection within a population. Studies in progress have developed methods to model polygenic selection in an admixture graph (representing of the historical divergences and admixture events in the human populations through time) to infer which branches are most likely to have experienced polygenic selection[55], and to model the relationship between variance in SNP effect and MAF to detect signatures of negative selection on variants associated with complex traits[56,57].

In summary, we proposed a robust statistical approach to test whether SNPs associated with a complex trait of interest are more differentiated across worldwide populations than MAF-matched and LD-matched control SNPs, and used the results to infer whether the trait-associated genetic variants have undergone natural selection. Our simulations indicated that the test statistics of the proposed approach were not inflated under the null model of random drift. Using this approach, we identified that for height, WHRadjBMI, and SCZ, the trait-associated alleles were differentiated significantly more than the matched control, in directions consistent with those of phenotypic differentiation. We showed that the results were robust to the potential biases in ascertaining the trait-associated SNPs (e.g., population stratification in EUR). These results support our hypothesis that the observed phenotypic differentiation among worldwide populations is (at least partly) genetic and a consequence of natural selection on the trait-associated variants because of selection on the trait or through their pleiotropic effects on fitness since the divergence of these populations[16]. Our findings further suggest that natural selection has also driven differentiation in LD among populations at genomic loci associated with height and SCZ. These findings expand our understanding of the role of natural selection in shaping the genetic architecture of complex traits in human populations. The methods developed in this study are general and applicable to other complex traits, including endophenotypes, such as gene expression and DNA methylation.

## Methods

**Data and quality control (QC).** The 1000G samples used in this study comprised individuals of EUR ($n = 503$), AFR ($n = 661$) and EAS origins ($n = 504$) (Supplementary Table 2). SNPs with MAF <0.01 and Hardy–Weinberg equilibrium (HWE) $P<10^{-6}$ were removed from the genotype data for each population, respectively, resulting in 6,160,018 SNPs in common across the three populations. We used GCTA[58] to construct the genetic relationship matrix (GRM) using the SNPs present in HapMap phase 3 project (HapMap3; $m = \sim 1.2$ million SNPs) and generate a set of unrelated individuals at a relatedness threshold of 0.05 in each population, resulting in 494 unrelated EUR, 591 unrelated AFR and 491 unrelated EAS (Supplementary Table 2).

There were 60,586 EUR, 3826 AFR, and 5188 EAS genotyped on Affymetrix Axiom arrays in the GERA data. The SNP genotype data were cleaned according to the following QC criteria: sample/SNP call rate <98%, MAF<0.01, and HWE test $P<10^{-6}$. After QC, the SNP genotypes were imputed to the 1000G (phase 1) reference panels using IMPUTE2[59] (Supplementary Table 6). The imputed SNPs with imputation INFO scores <0.3, MAF<0.01, or HWE $P<10^{-6}$ in any of the populations (AFR, EAS, and EUR) were removed (Supplementary Table 6), resulting in ~5.8 million remaining SNPs in common across the three populations. We performed a principal component analysis[60] (PCA) in a combined GERA and 1000G sample using 820,460 HapMap3 SNPs. For a particular population (e.g., GERA-EUR), any GERA individuals who were more than 3 s.d. away from the mean of the corresponding 1000G population (e.g., 1000G-EUR) were removed (Supplementary Fig. 16). We further calculated the GRM for each GERA population using the HapMap3 SNPs ($m = 820,460$), and removed one of each pair of individuals with estimated genetic relatedness >0.05. This resulted in 53,629 unrelated EUR, 1099 unrelated AFR and 3365 unrelated EAS. To avoid heterogeneity in the subsequent analyses, we harmonized the sample sizes by randomly sampling 1,099 individuals from the EUR and EAS samples (Supplementary Table 2). In summary, ~6.1 million SNPs in 1000G and ~5.8 million SNPs in GERA after QC were used in the subsequent analyses.

We used the LD-based clumping approach in PLINK[29] to select trait-associated SNPs from GWAS summary data. The clumping approach filters out SNPs with $P$-values larger than a specific threshold, clusters the remaining SNPs by LD and physical distance between SNPs, and selects the top associated SNP from each clump. We used an LD $r^2$ threshold of 0.01 and a distance threshold of 1 Mb to ensure that all the selected trait-associated SNPs were nearly independent. The Atherosclerosis Risk in Communities (ARIC) data set (~8.8 million 1000G-imputed SNPs and 7703 unrelated European Americans) was used as the reference to compute LD $r^2$ between SNPs. The ARIC sample was genotyped on Affymetrix 6.0 arrays. Genotype QC and imputation have been detailed elsewhere[61] (Supplementary Table 6).

**$F_{ST}$ enrichment test.** Similar to the approach used in Zhang et al.[30], we compared the mean $F_{ST}$ value of the trait-associated SNPs with that of the control SNPs with MAF and LD score matched. First, we divided all the SNPs (in either 1000G or HapMap2 depending on the SNPs used in GWAS for the trait) into 20 MAF bins from 0 to 0.5 with an increment of 0.025 (excluding the SNPs with MAF < 0.01). Each of the MAF bins was further grouped into 20 bins according to the 20 quantiles of LD score distribution. The MAF and LD values were computed from the EUR samples in GERA or 1000G described above. Second, we allocated the

trait-associated SNPs to the MAF and LD stratified bins, randomly sampled a matched number of "control" SNPs from each bin, computed a mean $F_{ST}$ value for the control SNPs sampled from all bins, and repeated this process 10,000 times to generate a distribution of mean $F_{ST}$ under drift (approximately normally distributed; see Fig. 1). Third, a P-value was computed from a two-tailed test by comparing the observed mean $F_{ST}$ value for the associated SNPs against the null distribution quantified by the control SNPs, assuming normality of the null distribution.

**Simulation based on real GWAS genotype data.** To verify whether the test statistics used in the $F_{ST}$ enrichment analyses are well calibrated, we used GCTA[58] to perform GWAS simulations based on the 1000G-imputed genotypes of 53,629 unrelated European Americans in GERA after QC. Two SNP panels (1000G and HapMap2) were respectively used to mimic those used in the real data to sample "causal variants" (GWAS for EAY, AD, CAD, and SCZ were based on 1000G and those for the other traits such as height and BMI were based on HapMap2) (Supplementary Tables 1 and 3). First, we simulated 100 quantitative traits under the null hypothesis (i.e., genetic drift) with heritability ($h^2$) of 0.5 based on 1000 causal variants randomly sampled from a SNP panel for each trait (1000G or HapMap2). The trait phenotype was simulated based on an additive genetic model, i.e.,

$$y = g + e = \sum_{i}^{m} w_i u_i + e \quad (1)$$

where $y$ is the phenotype, $m$ is the number of causal variants, $w_i$ is the standardized genotype of the $i$-th causal variant with its effect $u_i$ drawn from $N(0, 1)$, and $e$ is the residual generated from $N\left[0, \text{var}(g)\left(\frac{1}{h^2} - 1\right)\right]$. To demonstrate the statistical power under the alternative hypothesis (i.e., natural selection), we sampled 1000 causal variants for each trait from the top 50% of the $F_{ST}$ distribution of the 1000G (median $F_{ST}$ of 0.084) or HapMap2 (median $F_{ST}$ of 0.093) SNPs, where the $F_{ST}$ values were computed from the three populations in 1000G. Second, we performed a GWAS analysis for each trait (under null or alternative hypothesis) with the first 10 PCs fitted as covariates to control for population stratification. Independent trait-associated SNPs were selected by PLINK-clumping at two P-value thresholds ($5 \times 10^{-8}$ and $5 \times 10^{-6}$) alongside a LD $r^2$ threshold of 0.01 and window size of 1 Mb (Supplementary Table 3). We included a less stringent threshold (i.e., $5 \times 10^{-6}$) in the simulation to test whether the increased number of true positives passing this lower threshold could outweigh the increased number of false positives, which might result in an increase in power for the $F_{ST}$ enrichment test. Finally, we performed the $F_{ST}$ enrichment test for each trait to see if there is inflation under the null hypothesis and whether there is a difference in statistical power between the two P-value thresholds under the alternative hypothesis. Note that, for consistency, we used the same SNP panel to simulate the causal variants and to sample control SNPs. Four sets of results were generated from the combinations of two P-value thresholds ($5 \times 10^{-8}$ and $5 \times 10^{-6}$) and two SNP panels (1000G and HapMap2) (Supplementary Table 3).

We further performed simulations under the null hypothesis in three additional scenarios with 1) larger proportion of lower-MAF causal variant (1000 random causal variants + 500 causal variants with MAF < 0.1; A and B in Supplementary Table 4), 2) a lower level of heritability (i.e., $h^2 = 0.2$; C and D in Supplementary Table 4), and 3) two additional causal variants sampled from a 1-Mb flanking region of each primary causal variant (3000 causal variants in total). We also performed the $F_{ST}$ enrichment analysis of the original simulation data (see above) and the real GWAS summary data in two additional scenarios, i.e., 1) matching the control SNPs with the trait-associated SNPs by MAF (computed from 1000G-EUR) only, and 2) matching the control SNPs by both MAF and LD score computed from 1000G-AFR. The inflation of test statistics under the null hypothesis and the statistical power under the alternative hypothesis were illustrated by quantile-quantile (QQ) plots.

**Direction of genetic differentiation.** The analysis below uses a similar method introduced in Robinson et al. to quantify the population genetic differentiation of a complex trait[4]. The PRS of an individual is computed as $\hat{g} = \sum_{l}^{m} x_l \hat{b}_l$, where $m$ is the number of SNPs used to create the PRS, $x$ represents the SNP genotype (coded as 0, 1, or 2) and $\hat{b}$ is the estimate of SNP effect from the GWAS summary data. To investigate the direction of genetic differentiation among populations, we fitted a linear model

$$\hat{g}_i = \mu + v_j + e_i \quad (2)$$

where $\hat{g}_i$ represents the standardized PRS calculated from the trait-associated SNPs (clumping $P < 5 \times 10^{-6}$) for each of the unrelated individuals $i$ in either the 1000G or GERA samples; $\mu$ is the mean term; $v_j$ is the deviation of the mean PRS of population $j$ from $\mu$; and $e_i$ represents the residual. This method provides the estimate of the deviation (in s.d. units) of the mean PRS of a population from the overall mean. In data analysis, we also applied this method to estimate $v_j$ for

control SNPs (10,000 random sets for each trait in each population) to demonstrate the variability of $\hat{v}_j$ under drift.

**LD variation among populations.** Similar to the $F_{ST}$, we created LDCV for each SNP to measure LD variation among the AFR, EAS and EUR populations. We first calculated LD score of each SNP as the sum of the LD $r^2$ between the focal SNP and all the flanking SNPs (including the focal SNP itself) within a 10-Mb window. SNPs with LD $r^2$ values < 0.01 were excluded from the calculation to avoid chance correlations between SNPs. Each SNP obtained three LD scores estimated in the unrelated individuals from each of the three populations. We computed LDCV of a SNP as the ratio of the s.d. of the three LD scores to the mean in GERA and 1000G, respectively.

**Forward simulation.** We simulated two independent 10-Mb segments using SLiM[54] where new mutations occurred with 5% probability to be deleterious for fitness and 95% probability to be neutral on the first segment and 100% probability to be neutral on the second segment. The deleterious mutations were under negative selection with a selection coefficient of −0.01. The mutation rate was set to be $2.36 \times 10^{-8}$ (ref. [62]). The population samples were generated based on a commonly used demographic model[63], mimicking the "Out-of-Africa" event allowing migration between populations and population expansion. We started the simulation with 7310 individuals. After 58,000 generations as suggested in Gravel et al.[63], we obtained 34,039 Europeans and 14,474 Africans along with ~260,000 variants segregating in Europeans and ~225,000 variants in Africans. We sampled 5000 individuals from each population, extracted common variants (MAF > 0.01) and calculated the $F_{ST}$ (or LDCV) values of the variants in common between the two populations. We conducted the simulation with 30 independent replicates. The average number of common variants across the 30 replicates was 81,055 in Africans and 43,715 in Europeans with 29,710 variants in common. We then compared the mean $F_{ST}$ (or LDCV) value of the neutral variants on segment #2 with that of the non-deleterious variants with matching MAF and LD on segment #1 (some of which were under background selection because of the LD with deleterious mutations).

**URLs.** GCTA: http://cnsgenomics.com/software/gcta
SLiM: https://messerlab.org/slim
PLINK: https://www.cog-genomics.org/plink2
ARIC data: https://www.ncbi.nlm.nih.gov/projects/gap/cgi-bin/study.cgi?study_id=phs000090.v4.p1
GERA data: https://www.ncbi.nlm.nih.gov/projects/gap/cgi-bin/study.cgi?study_id=phs000674.v2.p2
GWAS summary data for Height, BMI, and WHRadjBMI: https://www.broadinstitute.org/collaboration/giant/index.php/GIANT_consortium_data_files
HDL and LDL: http://csg.sph.umich.edu/abecasis/public/lipids2013/
EAY: http://www.thessgac.org/data
AD: http://web.pasteur-lille.fr/en/recherche/u744/igap/igap_download.php
CAD (coronary artery disease): http://www.cardiogramplusc4d.org/data-downloads/
SCZ: https://www.med.unc.edu/pgc/downloads
T2D: http://diagram-consortium.org/downloads.html

**Data availability.** All the data used in this study were obtained from the public domain (see the URLs above).

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

## Acknowledgements

This research was supported by the Australian National Health and Medical Research Council (1107258, 1078037, 1103418, and 1113400), the Australian Research Council (DP160101343), the US National Institutes of Health (MH100141 and MH077139), and the Sylvia and Charles Viertel Charitable Foundation (Senior Medical Research Fellowship). This study uses data from dbGaP (accessions: phs000090 and phs000674). A full list of acknowledgments to these data sets can be found in the Supplementary Note.

## Author contributions

J.Y. conceived and designed the study. J.G. performed simulations and statistical analyses under the assistance and guidance from Y.W., Z.Z., Z.L.Z., M.T., J.Z., M.R.R., P.M.V.,

and J.Y. J.G. and J.Y. wrote the manuscript with the participation of all authors. All authors reviewed and approved the final manuscript.

## Additional information

**Competing interests:** The authors declare no competing interests.

