## [Peer Review File · Nature Communications]

Reviewers' comments:

Reviewer #1 (Remarks to the Author):

This paper shows that SNP that are associated with traits (height, BMI..) in GWAS are more differentiated between populations than 'random' SNP, and that the differences in genetic value predicted from GWAS tend to be in the same direction as for the actual phenotype. This is a very interesting study, and may eventually deserve publication. However, there seem to be two difficulties: it is not clear that the null distribution of F_{st} , obtained by simulation, is correct, and it cannot be argued that these observations are evidence for selection directly on the trait. Both these difficulties need to be resolved.

The description of the simulated null model is quite cryptic; I am not convinced that it properly takes account of haplotype structure. From the section around line 357 it seems that control SNP are used to calculate the null distribution of F_{st} . However, this does not take account of haplotype structure, which can greatly inflate the variance of F_{st} . The method is tested against simulation, but it is unclear what these simulations assume, and whether they reproduce the actual structure seen in the data. Note that variance in F_{st} can be greatly inflated even by weak correlations, which might not be corrected for by 'clumping' strongly associated SNP. This is not at all an easy problem to resolve - but as it stands, I am not convinced.

The second issue is more one of interpretation. Mostly, the authors are careful only to say that "trait-associated SNP are more differentiated", but there are several places where they suggest evidence that the traits are selected - which is not the case. First, it is unsurprising that PRS go in the same direction as the phenotype - that just shows that GWAS are predictive across populations. The question is rather, how to interpret greater differentiation and LD in trait associated SNP (if that is significant - see above). A plausible view is that trait variation is due to alleles that have pleiotropic effects on fitness, and that alleles with larger effects on the trait tend to be more differentiated via these pleiotropic effects - not because of selection on the traits, or even on correlated traits. This argument on real vs apparent selection goes back to the early 1990s. Actually, I think it very likely that height (say) does directly affect fitness - but I don't think that the evidence presented here strengthens that view. This problem needs to be stated explicitly, so that the reader does not get the wrong impression.

- The idea that selected loci will show a different F_{st} goes back to Lewontin and Krakauer, who should be cited.

30 - "differences in LD" needs to be explained a bit more, even for the abstract.

33 "Our results support .." - at best they are consistent with the hypothesis that selection has shaped phenotypic differentiation" - this sentence could be taken to imply that there is evidence for selection on the traits themselves.

84 sweep->sweeps

97-99 - Just using $MAF > 0.01$ does not correct for effects of frequency - though it seems that a finer scale correction was done, that is buried in the methods.

101 - "quantify"->"quality"

110 - "clumping" needs to be explained.

192 - As argued above, this study does not show that differences are "driven by selection" (at least, in the most obvious sense). I suppose one could still say that the study "sought to" show this, but that is misleading....

225 and later GWASs->GWAS

241 delete "history"

244 controls->control

245 comma after random SNPs
263 delete comma after showed
269 from->of
292 again, one cannot tell whether the trait has undergone differentiation via selection.
300 - pleiotropy refers to effects of genes, not to effects of the trait on fitness.
355 "Second, we"
358 It is not at all clear that this is the proper null distribution (see above)
375 - Wj=hat is GCTA-simu doing? This is utterly unclear. Does it generate a haplotype structure that is similar to that in the data?
383 "Four sets of results"

Nick Barton

Reviewer #2 (Remarks to the Author):

Guo et al. test GWAS associated variants for ten traits for polygenic selection. They find that associated SNPs for three traits are significantly more differentiated in allele-frequencies between populations than expected by chance. The differences in allele frequencies go in a direction that has some similarity to differences in the measurements of these traits between the populations. The authors also find differences in LD between populations that exceed those observed in control variants.

The study is generally well described. However, I was fairly confused by the line of argument in the paper and have some questions for the authors:

First, that the frequency of trait-associated variants in populations resemble the frequency of traits is a sign that GWAS does actually pick up truly associated variants and that these variants have some shared functionality in different populations. I do not see how this is in any way related to polygenic adaptation and find this line of argument confusing.

Second, it is not clear to me how the authors can exclude background selection as explanation for the F_{st} signal between populations. I do not think that matching LD and MAF would take care of this issue. Matching LD and MAF at the same time may also mask signals of selection, since LD is expected to be different for selected compared to non-selected variants when controlling for frequency. That the LD and MAF matching is carried out in only the European population worries me, too.

Third, the last analysis of the variance of LD over populations (comparing between trait-associated and control SNPs) is confusing to me. Are the control SNPs not already LD matched? What scenario are the authors suggesting for LD-variance increasing polygenic selection: That different loci are selected in different populations or that different alleles are selected? Why are the F_{st} outliers not also LD outliers if the latter process is the model?

There seem to be two related papers in the biorxiv, one with a large overlap in authors and one with no overlap:

<https://www.biorxiv.org/content/early/2017/06/03/145755>

<https://www.biorxiv.org/content/early/2017/11/08/146043>

It would be great if the authors could comment on these, but I understand of course that unpublished work doesn't need to be discussed.

Typos:

- I101: *quality* control

- I158: ARFs -> AFRs

Reviewers' comments

We thank the two expert reviewers for their constructive comments, which have significantly improved our manuscript. We have responded to all the reviewers' comments point-by-point below in this document (in blue) and have highlighted all the relevant changes in yellow in the revised manuscript.

Reviewer #1 (Remarks to the Author):

This paper shows that SNP that are associated with traits (height, BMI..) in GWAS are more differentiated between populations than 'random' SNP, and that the differences in genetic value predicted from GWAS tend to be in the same direction as for the actual phenotype. This is a very interesting study, and may eventually deserve publication. However, there seem to be two difficulties: it is not clear that the null distribution of F_{ST} , obtained by simulation, is correct, and it cannot be argued that these observations are evidence for selection directly on the trait. Both these difficulties need to be resolved.

Re: We thank the reviewer for the positive remarks and have performed additional simulations and analyses to address the two major concerns below.

The description of the simulated null model is quite cryptic; I am not convinced that it properly takes account of haplotype structure. From the section around line 357 it seems that control SNP are used to calculate the null distribution of F_{ST} . However, this does not take account of haplotype structure, which can greatly inflate the variance of F_{ST} . The method is tested against simulation, but it is unclear what these simulations assume, and whether they reproduce the actual structure seen in the data. Note that variance in F_{ST} can be greatly inflated even by weak correlations, which might not be corrected for by 'clumping' strongly associated SNP. This is not at all an easy problem to resolve - but as it stands, I am not convinced.

1) Re "The description of the simulated null model is quite cryptic" and "it is unclear what these simulations assume, and whether they reproduce the actual structure seen in the data".

We have provided more details about the simulation (page 12) in the revised manuscript.

To reproduce the actual structure in the data, we performed simulation based on real GWAS genotype data. We sampled 1000 SNPs at random from the 1000G-imputed GERA-EUR data as causal variants, and generated quantitative traits based on an additive genetic model $y = g + e = \sum_i^m x_i b_i + e$, where y is the phenotype, m is the number of causal variants, x_i is the standardized genotype of the i -th causal variant with its effect b_i drawn from $N(0, 1)$, and e is the residual generated from $N[0, \text{var}(g) \left(\frac{1}{h^2} - 1 \right)]$ with h^2 being the heritability. We performed a GWAS analysis for each of the simulated trait in GERA-EUR, and selected the top associated SNPs with $P < 5e-8$ by the clumping approach in PLINK (see below or the revised manuscript for a detailed description of the clumping approach). We then tested whether the mean F_{ST} value of the top associated SNPs is significantly different from that of the control SNPs with matching MAF and LD scores, where the F_{ST} values were computed from the EUR, AFR and EAS samples in 1000G. We show in Supplementary Fig. 1 that the test-statistics of the F_{ST} enrichment analysis are not inflated, suggesting the false positive rate is well controlled in the absence of selection.

In the previous version of the manuscript, the simulated causal variants were included in the association analysis, which might not be realistic because in reality most causal variants are not genotyped. In the revised manuscript, we re-ran the F_{ST} enrichment analysis excluding the simulated causal variants, and still did not observe inflation in the test-statistics of the F_{ST} enrichment analysis (Supplementary Fig. 11).

2) Re “I am not convinced that it properly takes account of haplotype structure” and “variance in F_{ST} can be greatly inflated even by weak correlations”

We used the LD-based clumping approach in PLINK to select trait-associated SNPs from GWAS summary data. The clumping approach filters out SNPs with P -values larger than a specific threshold, clusters the remaining SNPs by LD r^2 and physical distance between SNPs, and selects the top associated SNP from each clump. We used an LD threshold of 0.01 and a distance threshold of 1 Mb to ensure that all the selected trait-associated SNPs were nearly independent. We have included these details in the revised manuscript (page 11).

We have investigated the issue regarding to the variance of F_{ST} using simulated data (i.e. 100 independent traits simulated based on 1,000 causal variants randomly sampled from the GERA-EUR genotypes; Supplementary Table 3 iii). We calculated the variance of F_{ST} values (i.e. $\text{var}(F_{ST})$) for the trait-associated SNPs and for a random set of control SNPs (with MAF and LD scores matched). We then compared the mean estimate of $\text{var}(F_{ST})$ over 100 simulated traits between the two SNP sets. We did not observe significant difference in $\text{var}(F_{ST})$ between the trait-associated SNPs selected from the clumping analysis and the control SNPs sampled at random (mean = 0.0118 with s.e.m. = 0.00012 for associated SNPs and mean = 0.0116 with s.e.m. = 0.00012 for control SNPs; $P_{\text{difference}} = 0.193$) (Supplementary Fig. 12a), consistent with the observation that there was no inflation in the F_{ST} enrichment test-statistics in the absence of selection (page 8).

To investigate the LD issue further, we conducted a simulation based on 1,000 common causal variants (MAF > 0.01) sampled at random (i.e. no selection) with 2 additional causal variants sampled from a 1-Mb region flanking each of primary causal variants (page 8 in Discussions and page 12 in Methods). The purpose of this design is to simulate multiple causal variants at a single locus and thereby introduce LD among the causal variants at each locus (Supplementary Fig. 8). We repeated the simulation 100 times and did not observe inflation of the F_{ST} enrichment test-statistics in the absence of selection (Supplementary Fig. 9).

We have included all the additional analyses described above in the revised manuscript.

The second issue is more one of interpretation. Mostly, the authors are careful only to say that “trait-associated SNP are more differentiated”, but there are several places where they suggest evidence that the traits are selected - which is not the case. First, it is unsurprising that PRS go in the same direction as the phenotype - that just shows that GWAS are predictive across populations. The question is rather, how to interpret greater differentiation and LD in trait associated SNP (if that is significant - see above). A plausible view is that trait variation is due to alleles that have pleiotropic effects on fitness, and that alleles with larger effects on the trait tend to be more differentiated via these pleiotropic effects - not because of selection on the traits, or even on correlated traits. This argument on real vs apparent selection goes back to the early 1990s. Actually, I think it very likely that height (say) does directly affect fitness - but I don't think that the evidence presented here strengthens that view. This problem needs to be stated explicitly, so that the reader does not get the wrong impression.

Re: We thank the reviewer for pointing out this.

We have gone through the whole manuscript carefully and revised all the statements that might read as if we indicate that the traits have been under selection. We have now made it clear that our results only support the hypothesis that the trait-associated genetic variants have undergone natural selection because of selection on the trait or pleiotropic effects of the trait-associated variants on fitness.

We agree with the reviewer's comment that “it is unsurprising that PRS go in the same direction as the phenotype - that just shows that GWAS are predictive across populations”, and have clarified in the revised manuscript that this consistency should not be viewed as the evidence of natural selection unless the genetic differentiation is more than expected under drift (page 4).

We also agree with the reviewer's comment that "trait variation is due to alleles that have pleiotropic effects on fitness, and that alleles with larger effects on the trait tend to be more differentiated via these pleiotropic effects - not because of selection on the traits", and have included a similar statement in the revised manuscript (page 6).

- The idea that selected loci will show a different F_{ST} goes back to Lewontin and Krakauer, who should be cited.

Re: Done.

30 - "differences in LD" needs to be explained a bit more, even for the abstract.

Re: Done (pages 1 and 3).

33 "Our results support .." - at best they are consistent with the hypothesis that selection has shaped phenotypic differentiation" - this sentence could be taken to imply that there is evidence for selection on the traits themselves.

Re: We have revised the sentence as (page 1):

"Our results support the hypothesis that natural selection has shaped the differentiation of frequencies and/or LD scores at SNPs associated with complex traits, such as height and schizophrenia, among worldwide populations."

97-99 - Just using $MAF > 0.01$ does not correct for effects of frequency - though it seems that a finer scale correction was done, that is buried in the methods.

Re: We focused our analyses on common variants (i.e. $MAF > 0.01$) because rare variants were not available in most GWAS summary data used in this study. We have clarified this in the revised manuscript (page 3 in Results).

We have also clarified how we corrected for the effects of frequency (page 7):

We matched the trait-associated SNPs with the control SNPs by MAF because F_{ST} is a function of MAF and heterogeneity in MAF between the two SNP sets will result in a difference in mean F_{ST} even in the absence of natural selection. We further matched the trait-associated SNPs with the control SNPs by LD score to avoid potential bias due to the ascertainment of trait-associated from EUR-based GWAS (see below for more discussion).

110 - "clumping" needs to be explained.

Re: The clumping approach filters out SNPs with p-values larger than a specific threshold, clusters the remaining SNPs by LD r^2 and physical distance between SNPs, and selects the top associated SNP from each clump. We used an LD threshold of 0.01 and a distance threshold of 1 Mb to ensure that all the selected trait-associated SNPs were nearly independent. We have included these details in the revised manuscript (page 11).

192 - As argued above, this study does not show that differences are "driven by selection" (at least, in the most obvious sense). I suppose one could still say that the study "sought to" show this, but that is misleading....

Re: We have revised the sentence as (page 6):

"This study sought to address whether natural selection has differentiated allele frequencies and LD scores of complex trait associated SNPs among worldwide populations more than expected under drift."

84 sweep->sweeps

101 - "quantify"->"quality"

225 and later GWASs->GWAS
241 delete "history"
244 controls->control
245 comma after random SNPs
263 delete comma after showed
269 from->of
355 "Second, we"
383 "Four sets of results"

Re: We thank the reviewer for picking up the typos and errors, all of which have been corrected in the revised manuscript.

292 again, one cannot tell whether the trait has undergone differentiation via selection.

Re: We have revised the sentence as (page 10):

"In summary, we proposed a robust statistical approach to test whether SNPs associated with a complex trait of interest are more differentiated across worldwide populations than MAF- and LD-matched control SNPs, and used the results to infer whether the trait-associated genetic variants have undergone natural selection."

300 - pleiotropy refers to effects of genes, not to effects of the trait on fitness.

Re: We have revised the sentence as (page 10):

"These results support our hypothesis that the observed phenotypic differentiation among worldwide populations is (at least partly) genetic and a consequence of different selection pressures on the trait-associated variants because of selection on the trait or through their pleiotropic effects on fitness since the divergence of these populations¹⁶"

358 It is not at all clear that this is the proper null distribution (see above)

Re: This is the process to generate the distribution of mean F_{ST} values under a drift model. We have revised the sentence as (page 11):

"Second, we allocated the trait-associated SNPs to the MAF and LD stratified bins, randomly sampled a matched number of "control" SNPs from each bin, computed a mean F_{ST} value for the control SNPs sampled from all bins, and repeated this process 10,000 times to generate a distribution of mean F_{ST} under drift (approximately normally distributed; see Fig. 1)."

We have also mentioned in the Discussion section (page 9) that one of the limitations of this study is that the F_{ST} enrichment test against this distribution is underpowered for highly polygenic traits because some of the control SNPs might be by chance in LD with the causal variants under a polygenic model (as demonstrated by the deviation of F_{TIA} differences in the control SNPs from the expected values for height and SCZ; Fig. 3). Fortunately, the loss of power was remedied by the use of data from studies with very large sample sizes (Supplementary Table 1).

375 - Wj=hat is GCTA-simu doing? This is utterly unclear. Does it generate a haplotype structure that is similar to that in the data?

Re: We have clarified this in the revised manuscript (page 12).

As mentioned above, to reproduce the actual structure in the data, we performed the simulation based on real GWAS genotype data. We sampled 1000 SNPs at random from the 1000G-imputed GERA-EUR data as causal variants, and generated quantitative traits based on an additive genetic model $y = g + e = \sum_i^m x_i b_i + e$ using the simulation function in GCTA (Yang et al. 2011 AJHG), where y is the phenotype, m is the number of causal variants, x_i is the standardized genotype of the i -th causal

variant with its effect b_i drawn from $N(0, 1)$, and e is the residual generated from $N[0, \text{var}(g) \left(\frac{1}{h^2} - 1\right)]$ with h^2 being the heritability.

Reviewer #2 (Remarks to the Author):

Guo et al. test GWAS associated variants for ten traits for polygenic selection. They find that associated SNPs for three traits are significantly more differentiated in allele-frequencies between populations than expected by chance. The differences in allele frequencies go in a direction that has some similarity to differences in the measurements of these traits between the populations. The authors also find differences in LD between populations that exceed those observed in control variants.

The study is generally well described. However, I was fairly confused by the line of argument in the paper and have some questions for the authors:

First, that the frequency of trait-associated variants in populations resemble the frequency of traits is a sign that GWAS does actually pick up truly associated variants and that these variants have some shared functionality in different populations. I do not see how this is in any way related to polygenic adaptation and find this line of argument confusing.

Re: We thank the reviewer for this comment.

We have clarified in the revised manuscript (page 4) that the consistency (or resemblance) in direction between genetic and phenotypic differentiations (e.g. the result from the PRS analysis shown in Fig. 2) should not be viewed as evidence for natural selection unless the genetic differentiation is more than expected under drift.

We used the F_{ST} and LDCV enrichment analyses to detect signatures of natural selection. F_{ST} is the Wright's fixation index that is commonly used measure the variation of the frequency of a SNP across populations, and LDCV is metric proposed in this study to measure the variation of the LD score of a SNP across populations. The aim of this study is to investigate whether the variation in allele frequency (measured by F_{ST}) or LD (measured by LDCV) at the trait-associated loci is on average larger than expected under a drift model, an idea that goes back to Lewontin and Krakauer (1973) as pointed out by reviewer #1.

We therefore tested whether the mean F_{ST} (or LDCV) of trait-associated SNPs is significantly higher than that of randomly sampled control SNPs. We matched the trait-associated SNPs with the control SNPs by MAF because F_{ST} is a function of MAF and heterogeneity in MAF between the two SNP sets will result in a difference in mean F_{ST} even in the absence of natural selection. We further matched the trait-associated SNPs with the control SNPs by LD score to avoid potential bias due to the ascertainment of trait-associated from European-based GWAS (see page 7 for discussion).

We have shown by simulation (page 8) that the F_{ST} enrichment method is robust regardless whether the control SNPs were matched with the trait-associated SNPs by MAF and LD score computed from Europeans, only by MAF computed from Europeans, or by both MAF and LD score computed from Africans (Supplementary Fig. 10).

Second, it is not clear to me how the authors can exclude background selection as explanation for the F_{ST} signal between populations. I do not think that matching LD and MAF would take care of this issue. Matching LD and MAF at the same time may also mask signals of selection, since LD is expected to be different for selected compared to non-selected variants when controlling for frequency. That the LD and MAF matching is carried out in only the European population worries me, too.

Re: We thank the reviewer for the comments.

1) Re “background selection”

We thank the reviewer for this comment. Our results seem to indicate that the excess of genetic differentiation at the trait-associated SNPs is a consequence of local selection (different alleles are favoured in different populations). However, we cannot rule out the possibility that the increase in F_{ST} (or LDCV) at the trait-associated SNPs is due to background selection (Charlesworth et al. 1997 Genetics), i.e. those SNPs are in LD with causal variants under negative selection. This is confirmed by forward simulation using SLiM (Messer 2013 Genetics) (Methods). We conducted the simulation to mimic the “Out of Africa” event based on a commonly used demographic model (Gravel et al. 2011 PNAS). We simulated two independent 10-Mb segments, one with 5% of the mutations being deleterious for fitness and 95% being neutral and the other with all the mutations being neutral. We simulated negative selection on the deleterious variants so that the neutral variants on segment #1 in LD with the deleterious variants were under background selection while all the variants on segment #2 were unaffected (see page 13 for more details of the simulation). Over 30 simulation replicates, the average number of neutral variants was 16,860 (s.e.m. = 155), significantly larger than that of the variants under background selection (mean = 12,808 and s.e.m. = 109). The mean of mean F_{ST} values for the neutral variants across 30 replicates was 0.117 (s.e.m. = 0.002), significantly lower than that of the variants under background selection (mean = 0.137 and s.e.m. = 0.002) (Supplementary Fig. 15a). The corresponding value for LDCV was 0.411 (s.e.m = 0.003), also significantly lower than that under background selection (mean = 0.44 and s.e.m = 0.004) (Supplementary Fig. 15b). These results show that background selection reduces genetic diversity and increases between-population differentiation at genetic variants in LD with the variants under negative selection. We have included these in the revised manuscript (pages 9 and 13; Supplementary Fig. 15).

2) Re “Matching LD and MAF at the same time may also mask signals of selection” and “That the LD and MAF matching is carried out in only the European population worries me”

As mentioned in the response above, we matched the trait-associated SNPs with the control SNPs by MAF because F_{ST} is a function of MAF and heterogeneity in MAF between the two SNP sets will result in a difference in mean F_{ST} even in the absence of natural selection. We further matched the trait-associated SNPs with the control SNPs by LD score to avoid potential bias due to the ascertainment of trait-associated from European-based GWAS (see page 7 for discussion).

We have shown by additional simulations (page 8) that the F_{ST} enrichment method is robust regardless whether the control SNPs were matched with the trait-associated SNPs by MAF and LD score computed from Europeans, only by MAF computed from Europeans, or by both MAF and LD score computed from Africans (Supplementary Fig. 10). We have also applied different strategies of matching control SNPs to the analyses of real data and observed little differences in results (Supplementary Table 5).

Third, the last analysis of the variance of LD over populations (comparing between trait-associated and control SNPs) is confusing to me. Are the control SNPs not already LD matched? What scenario are the authors suggesting for LD-variance increasing polygenic selection: That different loci are selected in different populations or that different alleles are selected? Why are the F_{ST} outliers not also LD outliers if the latter process is the model?

Re: In the LDCV enrichment analysis, the control SNPs were matched with the trait-associated SNPs by LD score in a population rather than the variation in LD across populations. This is similar to the F_{ST} enrichment analysis where we matched the control SNPs with the trait-associated SNPs by allele frequency and tested the difference in variation in allele frequency across populations (measured by F_{ST}) between the trait-associated SNPs and control SNPs.

Previous studies have shown that both strong and soft selective sweeps can alter LD between a genetic variant and its surrounding variants. If a trait-associated SNP has been under selection, one would expect to see a population differentiation of the LD of this SNP with its surrounding SNPs, more than expected under a drift model. We have clarified this in the manuscript (page 3). Because we tested the mean LDCV (or F_{ST}) across all the associated loci against that of the control SNPs, we could not

distinguish whether the population differentiation of mean LDCV (or F_{ST}) at the trait-associated variants (more than expected by drift) is due to natural selection on different loci, the same loci but different alleles, or the same alleles but different levels of selection pressure in different populations. We have discussed this in the revised manuscript (page 9).

There seem to be two related papers in the biorxiv, one with a large overlap in authors and one with no overlap:

<https://www.biorxiv.org/content/early/2017/06/03/145755>

<https://www.biorxiv.org/content/early/2017/11/08/146043>

It would be great if the authors could comment on these, but I understand of course that unpublished work doesn't need to be discussed.

Re: We thank the reviewer for pointing these two papers to us. We have commented both of them in the revised manuscript (page 9).

“We used the F_{ST} and LDCV enrichment analyses to assess the excess of population genetic differentiation at the trait-associated loci as a means to detect signature of natural selection. These analyses, however, cannot determine when the selection occurred in the history of human evolution and whether there are other types of natural selection within a population. Studies in progress have developed methods to model polygenic selection in an admixture graph (representing of the historical divergences and admixture events in the human populations through time) to infer which branches are most likely to have experienced polygenic selection (Racimo et al. 2017 bioRxiv), and to model the relationship between variance in SNP effect and MAF to detect signatures of negative selection on variants associated with complex traits (Zeng et al. 2017 bioRxiv; Schoech et al. 2017 bioRxiv).”

Typos:

- l101: *quality* control

- l158: ARFs -> AFRs

Re: We thank the reviewer for picking up the typos, both of which have been corrected in the revised manuscript.

REVIEWERS' COMMENTS:

Reviewer #1 (Remarks to the Author):

The authors have addressed all my previous concerns thoroughly, and I am now happy to see this work published. I am actually surprised (in a positive way) that the variance of F_{st} matches simulations, suggesting that my previous concern about LD was not justified. Overall, this is a nice study, carefully done.

Reviewer #2 (Remarks to the Author):

It is great that the authors carried out forward simulations to test background selection. Unfortunately, the authors find that background selection can explain the signals, and I see little reason to invoke selection on the traits as an alternative explanation.

Minor: The abstract currently makes a circular argument. You test for selection on trait-associated alleles using allele-frequency differentiation and LD scores. The last sentence says that your results support the hypothesis that selection shaped allele-frequency differences and LD-scores. If you use these signals to detect selection, then you cannot also claim that they are shaped by selection. You make that assumption already for your tests.

Response to Reviewers' comments

Reviewer #1 (Remarks to the Author):

The authors have addressed all my previous concerns thoroughly, and I am now happy to see this work published. I am actually surprised (in a positive way) that the variance of F_{st} matches simulations, suggesting that my previous concern about LD was not justified. Overall, this is a nice study, carefully done.

Reviewer #2 (Remarks to the Author):

It is great that the authors carried out forward simulations to test background selection. Unfortunately, the authors find that background selection can explain the signals, and I see little reason to invoke selection on the traits as an alternative explanation.

Minor: The abstract currently makes a circular argument. You test for selection on trait-associated alleles using allele-frequency differentiation and LD scores. The last sentence says that your results support the hypothesis that selection shaped allele-frequency differences and LD-scores. If you use these signals to detect selection, then you cannot also claim that they are shaped by selection. You make that assumption already for your tests.

Re: We thank the reviewer for this comment and have revised the last sentence as:
"Our results support the hypothesis that natural selection has shaped the genetic differentiation of complex traits, such as height and schizophrenia, among worldwide populations."